# On an Aggregated Estimate for Human Mobility Regularities through Movement Trends and Population Density

**DOI:** 10.3390/e26050398

**Published:** 2024-04-30

**Authors:** Fabio Vanni, David Lambert

**Affiliations:** 1Department of Economics, University of Insubria, 21100 Varese, Italy; 2Université Côte d’Azur, CNRS, GREDEG, 06103 Nice-Sophia Antipolis, France; 3Department of Physics, University of North Texas, Denton, TX 76205, USA; davidlambert2@my.unt.edu

**Keywords:** human mobility, collisional mathematical model, population density, economic trends, economic time series analysis

## Abstract

This article introduces an analytical framework that interprets individual measures of entropy-based mobility derived from mobile phone data. We explore and analyze two widely recognized entropy metrics: random entropy and uncorrelated Shannon entropy. These metrics are estimated through collective variables of human mobility, including movement trends and population density. By employing a collisional model, we establish statistical relationships between entropy measures and mobility variables. Furthermore, our research addresses three primary objectives: firstly, validating the model; secondly, exploring correlations between aggregated mobility and entropy measures in comparison to five economic indicators; and finally, demonstrating the utility of entropy measures. Specifically, we provide an effective population density estimate that offers a more realistic understanding of social interactions. This estimation takes into account both movement regularities and intensity, utilizing real-time data analysis conducted during the peak period of the COVID-19 pandemic.

## 1. Introduction

Human mobility and its regularity is an important and interdisciplinary research field and many theoretical studies have explored it at both individual and group levels [1,2,3].

Human mobility exhibits both random and regular patterns, reflecting a complex interplay of social interactions. Entropy offers a means to quantify these patterns [4,5,6], particularly in the context of understanding employees’ and consumers’ mobility behaviors. The economic significance of these insights is considerable, as mobility patterns serve as powerful predictors of consumer behavior and commercial activity [7,8]. The concept of entropy is a fundamental pillar within statistical physics [9,10] and has subsequently been applied across various disciplines such as biology, information theory, and economics. A particularly important application is in the realm of epidemiological studies, including the modeling of COVID-19 cases and the basic reproduction number, where entropy-based models and metrics have been introduced for global transmission to measure the impact and the temporal evolution of a pandemic event [11,12]. Our analytical framework aims to provide an interpretation of individual measures of entropy-based mobility as obtained from mobile phone data [13,14]. We introduce and analyze the well-known random entropy and uncorrelated Shannon entropy, and estimate them through collective variables of human mobility such as movement trends and population density. We adopt a collisional model that allows us to establish statistical relationships between entropy measures and mobility variables. In recent decades, multiple models have emerged to elucidate fundamental principles of human mobility, such as the gravity model, radiation model, and opportunities model [15,16], as well as the role of human mobility in the context of infectious disease spread [17,18,19,20], highlighting the importance of human interactions in understanding the evolution over the course of a pandemic.

We focus our data analysis on mobile phone data, utilizing measures such as radius of gyration and mobility entropy to characterize movement patterns. User location datasets can be used to study and model user mobility behaviors [21,22,23]. We utilized a database containing entropy and movement metrics at the U.S. state-level regions, which provided free access to the data throughout the year 2020. The time period of the early COVID-19 pandemic is particularly interesting as mobility restrictions and other social distancing interventions were implemented to mitigate the spread of the disease. These measures had significant impacts on economic activities, amplifying the effects of changes in mobility behaviors on social, economic, and demographic trends and patterns [24,25,26,27].

We define a novel approach to the measurement of population density, one of the most fundamental properties of urban areas. Many research works have looked at the relationship between the density, mobility, productivity, and urban development of a region(s) [28]. Population-weighted density captures density as perceived by a randomly chosen individual and is meant to measure average “experienced” density, as popularized in economics [29,30]. We propose an adjustment factor to population density defined in terms of the mobility patterns of a region. By using mobility and demographic data information, we observe how the proposed entropy-weighted population density suggests that people perceive interactions as more effective when there is frequent movement with low irregularity.

Finally, we investigate how mobility constraints influence employment, consumer choices, and market dynamics. Human movement patterns are strongly associated with regional socioeconomic indicators [31,32,33,34]. Mobile phone data can be used as a proxy to evaluate the density, activity, and social characteristics of a population [35,36,37]. The economic and social shock presented by abrupt or intense changes in human mobility and population density has reshaped the perceptions of individuals and organizations about work and occupations. Employment rate, for example, is known to be strongly related to aggregate measures of consumer spending [38,39], especially during times of crisis. The analysis of human mobility regularity during epidemic outbreaks such as COVID-19 is of crucial importance in social sciences and economics. This significance stems from the intricate interplay between aggregate demand and epidemic dynamics. The trajectory of the epidemic impacts consumer behavior, thereby influencing demand trends and the frequency of physical contact among individuals, ultimately affecting epidemic incidence [40,41,42]. Increased aggregate demand raises the contact rate, thereby heightening the risk of virus transmission. Conversely, escalating infection rates dampen aggregate demand due to reduced household spending. Similarly, a higher epidemic incidence suppresses aggregate demand, resulting in a decrease in the contact rate and a subsequent reduction in infections.

Our analysis sheds light on the role of entropy measures of human mobility in economic analyses, contributing to the literature on the mathematical modeling of consumer behaviors in relation to mobility regularities [43,44,45]. We provide an epidemiological model for the effective reproduction number in combination with entropy-based metrics of human mobility and we reveal relationships between economic indicators and mobility-based variables, underscoring how economic activity and human movement are intertwined.

## 2. Methods and Models

Entropy is a significant, widely used, and (above all) successful measure for quantifying in-homogeneity, impurity, complexity, uncertainty, or unpredictability. We are interested in the entropy of a person’s large-scale motion. This metric tracks how erratic or predictable a person’s movements are. Lower entropy implies higher predictability, meaning that an individual’s time spent at all locations is highly predictable. Conversely, high entropy implies that predicting the time an individual spends in some location(s) is difficult.

Random entropy measures the uncertainty of an individual’s next location, assuming that this individual’s movement is completely random amongst *L* possible locations, and is estimated from data as follows:(1)$$\begin{array}{cc} E[{\hat{S}}_{\mathrm{rand}}] & ={\langle \log L_{i}\rangle }_{N} \end{array}$$
which captures the predictability of each user by assuming that the person’s whereabouts are uniformly distributed among $L_{i}$ distinct locations.

Uncorrelated entropy, on the other hand, is based on Shannon entropy: if the individual’s movements among *N* possible locations follow a certain probability distribution, the uncorrelated entropy is then estimated from data as follows:(2)$$E\left[{\hat{S}}_{\mathrm{unc}}\right]=-{\langle \sum_{k=1}^{L_{i}}p_{k}\log p_{k}\rangle }_{N}$$
where $L_{i}$ is, again, the number of distinct locations by each individual, i=1…N, and where $p_{k}$ is the frequency of the user’s visit to their *k*-th location (*k* is the index of all locations that the user visits). Shannon entropy is high when an individual performs many different trips from a variety of origins and destinations; it is low when he performs a small number of recurring trips. The uncorrelated Shannon entropy takes into account the number of different locations visited as well as the proportion of time spent at each location. Since the entropy of the distribution of time spent at a location is no lower than that of a uniform distribution, one has that $E\left[{\hat{S}}_{\mathrm{unc}}\right]\leq E\left[{\hat{S}}_{\mathrm{rand}}\right]$, where the equality holds when the process is completely random.

**Definition** **1**(collisional assumption)**.**
*Let individuals be point particles in a container of area A that interact with the borders of the container and otherwise move freely. Let the container be of such a shape as to ensure the chaotic motion of the particles.*

We start with the analytical estimation of random entropy through a configurational approach:

**Proposition** **1**(Configurational Entropy)**.**
*Let A be an area in which N non-interacting individuals randomly move. Let δ=N/A be the population density. In the configurational case, the random entropy can be computed as follows:*
(3)$$\begin{array}{cc} S_{R} & =\ln\frac{\delta_{o}}{\delta } \end{array}$$
*where $\delta_{o}$ is a characteristic density (typically, N individuals per unit area).*

**Proof.** The proposition is proven in Appendix A. □

Note that configurational entropy depends only on the population density. The presumed area equidistribution of individuals implicit in the definition of $S_{R}$ reminds one of the random entropy $E[{\hat{S}}_{\mathrm{rand}}]$ in Equation (Equation 1).

Human mobility encompasses various activities, such as commuting to work, shopping, or socializing, which collectively contribute to the dynamic fabric of human interactions within a given community or region. So, it would be desirable to have an entropy that would include both velocity and density as independent variables.

**Proposition** **2**(Informational Entropy)**.**
*Let N be the number of individuals in an area. Let the state of the set of individuals be represented by a point in the 4N-dimensional phase space. The principle of maximum entropy yields the following:*
(4)$$\begin{array}{c} S_{U}=1+\ln\frac{\delta_{o}}{\delta }\frac{\mu^{2}}{\mu_{o}^{2}} \end{array}$$
*where μ is the mobility and $\mu_{o}$ is the minimal resolvable velocity.*

**Proof.** The proposition is proved in Appendix B. □

In a sense, $S_{U}$ is a better measure of entropy than $S_{R}$ since it takes into account the spatiotemporal patterns in mobility. They are equivalent (up to a constant) only if the mobility distribution is unchanging in space and time.

## 3. Results

In this section, we discuss three main objectives of the research. First, we provide evidence concerning the relationship between entropy as directly estimated and reported from datasets and the entropy calculated from mobility data using configurational and informational entropies. Second, we illustrate how to calculate a novel measure of the population density as “experienced” based on entropy. Third, we highlight statistical evidence about the relationship between mobility regularities and certain macroscopic economic indicators. Finally, we provide a real-time epidemiological estimation of the effective reproduction number in relation to mobility patterns of regularities.

### 3.1. Entropy in Human Mobility

We use data provided by the Camber Systems database [46], which reports the random and uncorrelated Shannon entropies at the length scale of meters. Real-time user locations are collected using the global positioning system (GPS), call detail record logs (CDR), and a wireless LAN (WLAN). Recording human activities can yield high-fidelity proxies of socioeconomic development and well-being. However, mobility data have their biases and limitations. For example, these data can be more representative of a younger and more affluent population while under-representing those living in rural areas.

We use data provided by the Camber Systems database [46], which offers estimates for random entropy, uncorrelated (Shannon) entropy, and the radius of gyration (RoG). The latter, RoG, measures how far an individual moves from its trajectory’s center of mass. We identify RoG as a measure of mobility μ.

The human mobility data were retrieved from [46], where entropy is used as a measure of the regularity of mobility patterns, and the radius of gyration serves as a measure of the quantity of mobility in terms of the distance traveled. Our study focuses exclusively on the USA, as it was the only freely accessible database offering pertinent information essential for our analysis. In Table 1, we provide a typical structure of the data sample: the first column indicates the fips code of the state analyzed, the second column presents the day of activity, the following columns report Entropy (mean and standard deviation), the radius of gyration (mean and standard deviation) together with the number of home devices used. The time period is about 1 year at a daily resolution. (The dataset utilized originates from smartphones, inherently limiting the inclusivity of the overall dataset. Additionally, to obtain a representation as close to the general population as feasible, the dataset was compared with census data to normalize it as effectively as possible. Moreover, recognizing that historical and contemporary racial disparities are evident in residential and occupational patterns, it is imperative to contextualize movement within this framework and refrain from utilizing this information to make decisions that could perpetuate or exacerbate such disparities.). We explain the economic data in Section 3.3.

We now describe the two branches that comprise human mobility in our collisional model: mobility (RoG) and population density.

Mobility is the average distance traveled by an individual. In the data, it is represented by the median radius of gyration in meters of devices that stayed in one location overnight. This metric provides a summary of travel that incorporates the number of trips and the distance of each trip. The radius of gyration, $r_{g}^{\left(u\right)}$, for the user, *u*, is calculated by first taking the root mean squared distance of a user’s location from their trajectory’s center of gravity averaged over a given time window. (The radius of gyration is defined as $r_{g}^{\left(u\right)}=\sqrt{\frac{1}{L_{i}^{\left(u\right)}}\sum_{i=1}^{L_{i}^{\left(u\right)}}{\left(r_{i}^{\left(u\right)}-{\bar{r}}^{\left(u\right)}\right)}^{2}}$ where $r_{i}^{\left(u\right)}$ represents the $i=1\ldots L_{i}^{\left(u\right)}$ positions recorded for the device *u* and ${\bar{r}}^{\left(u\right)}$ is the center of mass of the trajectory). It is interpreted as the characteristic distance traveled by the user during a day (note that the radius of gyration for purely diffusive motion obeys $r_{g}\left(t\right)∼\sqrt{t}$ [47,48]). The individual radius of gyration is different from the average travel distance because an individual moving in a comparatively confined space will have a small radius of gyration even if a large distance is covered. The aggregated radius of gyration, RoG, is reported for a group of devices at a geohash-7 granularity of approximately 153 m by 153 m; for every user, *u*, one generates their home region, *A*, as the region in which they spend the most time in their location set. Then, one aggregates this value across a population in a given region to provide an average and percentiles. This is the metric used in the Camber Systems [46] database. In practice, the radius of gyration represents a way to describe human mobility as an aggregated measure of human movements. It can be taken as a measure of mean velocity, μ, in our collisional model and $\mu_{o}$ is the displacement resolution, which determines the minimal movement that can be detected in a geohash area in a day.The second aspect of human mobility in our collisional model is the number of individuals that can interact in a region. Despite its simplicity, population density δ is quite difficult to estimate, since we should map the number of individuals in an effective region of interaction. Most economic agents are more concentrated in space than gross population density measures suggest. For these reasons, informational entropy is just an approximation of the uncorrelated entropy since we cannot know the exact value of the urban population density. As a consequence, we cannot directly use the population density reported by the U.S. Census Bureau [49], which is calculated in terms of the area of a geographic region rather than the effective area in which people interact. We use the population-weighted density of [50] from Table 2: it represents the population density that the average person experiences [51]. It is often more representative of the effective interaction region standard census estimate [52].

**Table 2 entropy-26-00398-t002:** Data repositories at the state level in the U.S.

*Human Mobility Data*
Camber Systems Social Distancing Reporter [46]—*Entropy, RoG and Devices* (*daily*)
U.S. Census Bureau [49]—*Population density* (*annual*)
World Pop Hub [50]—*Population-weighted density* (*annual*)
*Economic Data*
Economic Tracker [53]—*Employment, Consumer Spending, and Firm Revenue* (*daily*)
U.S. Energy Information Administration [54]—*Energy Demand and Production* (*daily*)
Federal Reserve Bank of Philadelphia [55]—*Coincidence index* (*monthly*)

As a consequence, while human movement trend data, μ, are reliable mobility predictors for entropy, the population density, δ, cannot be measured with the same precision and frequency as movement trends.

We show in Figure 1 and Figure 2 the various measures of entropy in certain U.S. states. We notice that the estimated uncorrelated entropy $E[{\hat{S}}_{\mathrm{unc}}]$ is well approximated by the informational entropy $S_{U}$ expressed via human mobility variables. On the other hand, the random entropy $E[{\hat{S}}_{\mathrm{rand}}]$ is only roughly approximated by the configurational entropy since this relies on stronger assumptions and on the estimate of population density only, which is a quite crude measure. Nevertheless, it follows a similar trend in states where the population is more evenly distributed.

However, in every case reported, we noticed a change after the beginning of March 2020: random entropy has increased while uncorrelated entropy exhibits the opposite trend. A possible interpretation is that, after the COVID-19 outbreak, individuals increased the number of stops (visited more distinct locations), but at the same time, they moved less uniformly among different locations they had to stop at. This possibly reflects the fact that people kept moving but spent more time in important places like essential workplaces, and less time visiting non-necessary locations (e.g., bars, restaurants, etc.…). Over the course of a year, the uncorrelated Shannon entropy has recovered its original value, while the random Shannon entropy has maintained its new mean value. This suggests that the density of active individuals has diminished, possibly because of social and economic downturns and the reduction in production activities and consumption.

Finally, we would like to point out that we noticed disparities between workdays and weekends in human mobility patterns. Aggregated data reveal peak activity and mobility irregularity during weekdays, with notably reduced movements observed on weekends, corroborating findings from previous studies [56,57,58]. Specifically, pre-COVID-19, weekdays exhibited higher levels of aggregate mobility (RoG) compared to the pandemic period, while weekends similarly displayed higher mobility levels before COVID-19. The analysis of entropy in mobility patterns shows an opposite trend—entropy is lower on weekdays before the pandemic compared to the COVID-19 period, and a similar trend is observed for weekends. We observed that a high radius of gyration (RoG) coupled with low entropy might suggest a population that must travel significant distances for work or essential errands, yet predominantly remains at home otherwise. Conversely, a low RoG paired with high entropy might signify a densely populated area where individuals predominantly stay close to home but exhibit increased movement within their neighborhood. Both scenarios offer valuable insights requiring distinct intervention strategies. However, a more detailed and robust statistical analysis is required to comprehensively describe these phenomena, leaving room for further exploration in future research.

### 3.2. Population Density Estimate

The ordinary gross population density is defined as population divided by (land) area. This is a flawed measure since it deals with large geographic entities such as counties, states, and countries rather than smaller structured regions such as cities. As an economic measure, population density needs to accommodate the fact that most economic agents live in a more concentrated space than gross population density measures suggest. The two basic issues with average population density calculations are the arbitrariness of defining borders and the fact that average population density focuses on the density of the average plot of land, not the density observed by the average person.

There are several ways to correct the arbitrariness of defining borders, territorial characteristics, and people distribution in a region. The focus is to produce a sort of “perceived” population density. For example, the population-weighted density, proposed by [51,59], is a family of methods that weigh the value of density by their corresponding population size in the aggregation process. In particular, to gain a perspective on the densities at which people live, population-weighted density is derived from the densities of all the census tracts included within the boundary of the Core-based statistical area. In the specific case of the United States, we used the regular population density [49] and other weighted population density metrics at the sub-national (state) level for the year 2020, as retrieved via the WorldPop dataset [50]. Following the Shannon (uncorrelated) entropy from phone data [46], the perceived population density is as follows:(5)$$\begin{array}{cc} {\hat{d}}_{\mathrm{ew}} & =d_{0}\frac{\mu^{2}}{\mu_{o}^{2}}\exp\left\{1-E\left[{\hat{S}}_{\mathrm{unc}}\right]\right\} \end{array}$$
where we interpret $d_{0}$ as the number of individuals that one can count in a random geohash, which is the same as the gross density from the census bureau, i.e., the number of people per ${\mathrm{km}}^{2}$ of land area. In Table 3, we show the various adjustment criteria for the census population density. The first column shows the typical measure of population density as a number of persons divided by the nominal surface area of the state. The second column represents the two entropy-weighted population densities as computed by Equation (Equation 5). The last two columns report two other popular approaches to adjust and weight the population density. The first uses population divided by land area adjusted for geographic characteristics; see [60]. The last column represents the population-weighted density based on the weighted median, as suggested by [51], which is more strongly related to the size of the urban area.

In Figure 3, we show two population density maps: the gross population density and the Shannon entropy-weighted population density. The latter differs from the former by a correction factor, which takes into account both the amount of movement and the diversity of mobility. For a similar surface area and population, the experienced density is higher for regions where people tend to move more with high regularity (low entropy). The correction factor for random entropy only takes into account movement regularity. So, the entropy-weighted population density indicates that individuals perceive the interaction to be more effective if people move a lot with a low degree of confusion, i.e., in areas with irregular movements, individuals may encounter others less frequently, leading to a perception of lower population density. Conversely, in areas with more regular movements, encounters with others may be more frequent, leading to a perception of higher population density. Moreover, in areas with more irregular movements, individuals may have less predictable spatial relationships with others, which can decrease their perception of population density [61,62,63].

However, our estimate must be used with caution since it depends on the quality of its components and the assumptions behind the model that establish the relationship between mobility variables. More work is required to provide a more systematic study of the impact of different ways to define the average velocity, μ, which we denote here as the radius of gyration. Other alternatives are possible, such as the mean square displacement and the straight-line distance, which define different types of movement trends. Moreover, other definitions of entropy are also possible; for example, the real entropy is a metric that not only uses the frequency of visitation but also considers the order and the time spent in visited locations, thus capturing more complete spatiotemporal features of mobility patterns [3,4,64]. The take-home message is that—even if the actual population density remains constant—the perception of crowding can vary based on the intensity and irregularity of movements.

### 3.3. Economic Activity, Mobility Patterns, and Epidemiological Evolution

In this final section, we aim to highlight several stylized facts and empirical evidence concerning patterns of mobility and various socioeconomic indicators, such as short-term regional income, employment rates, and other socioeconomic factors [65,66,67]. We examine the correlations between aggregated mobility and entropic measures vis-à-vis five socioeconomic indicators. As a case study, we selected daily and monthly regional economic indicators that have already been analyzed in research on the short-term impact of the COVID-19 epidemic on the economy [25,68,69]. We used five economic indicators: employment, consumer spending, electricity production, firms’ revenues, and the coincidence index.

In Figure 4, we illustrate that the economic indicators shown in the plot appear to be better aligned with entropy rather than mobility, especially for U.S. states like New York and others where entropy and mobility exhibit opposite trends at the beginning of 2021. In states such as California, where the two mobility indicators show similar trends, the correlations between mobility and economic indexes are more indistinguishable.

At this point, we can assess the statistical measures that determine the association of these indicators with mobility and entropy variables for each state. We accomplish this by measuring the linear relationship through the correlation coefficient, thereby highlighting the aggregated extent to which each of the economic time series moves together with mobility and entropy. We demonstrate that some of the economic series are more strongly correlated with entropy compared to mobility alone. We attribute this to the varying information content in each economic variable and to the fact that entropy embodies more complex information than just mobility. While this may be advantageous for certain economic indicators, it may not hold true for others. In particular, we investigate the relationship between each of the selected economic indexes and the mobility and entropy variables for each of the 50 U.S. states, as reported in Figure 5 and Figure 6.

We observe in that entropy is systematically highly correlated with employment and the coincident index compared to mobility in almost all states. Conversely, mobility (such as RoG) shows a higher correlation with energy demand. As for the impacts on the labor market, employment rates in the United States fell dramatically during the first months of 2020 as the repercussions of the COVID-19 pandemic reverberated through the labor market. However, the pandemic-related economic pause and lockdown differently affected the employment opportunities of individuals working in different sectors. In particular, regions with economies relying on the movement of people (such as tourism) faced substantially higher unemployment at the end of 2020 than regions with core industries based on the movement of information. Population mobility is closely related to consumer decisions regarding what to buy, how much to buy, and when to buy among many goods and services. Consumers not only satisfy their own needs but also determine the quantity and types of goods and services ultimately produced. The production of these goods and services creates jobs in all sectors of the economy. From a consumer spending perspective, we selected only high-frequency data related to electricity demand. This indicator measures how much electricity each end-use sector consumes and the varying effects of COVID-19 mitigation efforts on the sectors. We used data on the demand for electricity as reported by the International Energy Agency (IEA) [54] of regional electricity production in megawatt-hour units. Ultimately, we selected the monthly coincident index for each of the 50 states as produced by the Federal Reserve Bank of Philadelphia [55]. The coincident indexes combine four state-level indicators to summarize the current economic conditions in a single statistic. These indexes are monthly indicators of economic activity for each of the 50 U.S. states, based on a composite of four widely available data series on state conditions: total non-farm payroll employment, the unemployment rate, average hours worked in manufacturing, and real wages/salary disbursements.

We summarize all the results above in terms of an overall analysis as reported in Table 4, where we compute the median correlations and their confidence intervals. We observe that entropy shows a significantly higher degree of correlation with employment and the coincidence index than mobility. This may be due to the more complex nature of entropy, which accounts for mobility along with other social distance measures as indicators of the regularity of location patterns. Meanwhile, mobility is more strongly correlated with energy demand, which can be attributed to the fact that electricity consumption is more sensitive to changes in the movements of individuals rather than some sort of regularity of those movements.

Building upon the empirical findings, we can lay the foundation for a mathematical model that links epidemic trends to human mobility and social distancing behaviors. These factors, in turn, both influence and are influenced by social and economic activities. In this context, it is important to note that informational entropy serves as a crucial variable in determining the effective reproduction number of an epidemic. The reproduction number, $R_{t}$, has been estimated via a renewal epidemic model as developed in [20,71]. The human mobility regularity pattern influences the epidemiological evolution as follows:

**Proposition** **3.**
*The evolution of the effective reproduction number can be written in terms of the uncorrelated Shannon entropy, S, through the following delayed equation:*

(6)
$$R_{t+\tau }=s_{t}\gamma_{t}\;R_{0}\;e^{\Delta S_{t}}$$

*where τ is a delay variable that accounts for the typical time it takes to observe a newly generated positive test, $R_{0}$ denotes the basic reproduction number, $s_{t}$ represents the fraction of susceptible individuals with respect to the initial time, t=0, $\gamma_{t}$ represents the infection transmissibility factor, and finally, $\Delta S_{t}=\frac{1}{2}\left(S_{t}-S_{0}\right)$ is the Shannon entropy (half) difference, with respect to its initial value $S_{0}$ at time t=0.*


**Proof.** The proposition is proven in Appendix C. □

From the previous proposition, we can see that the condition, $\Delta S_{t}>0$, will result in a positive contribution of entropy to an increase in the effective reproduction number. Conversely, the condition, $\Delta S_{t}<0$, will contribute to a decrease in the effective reproduction number. If $\Delta S_{t}=0$, the value of the effective reproduction will change uniquely in relation to the susceptible population and infection transmissibility. As a consequence, keeping all other variables constant, the more regular human movement becomes, the lower the reproduction number.

At this point, it is important to further explore the implications of Proposition 3 to establish an analytical framework based on the collisional model in Definition 1 to estimate cases or deaths resulting from the COVID-19 pandemic. We can estimate the daily number of cases by employing a Poisson process, following the renewal in Equation [20,72]:(7)$${\hat{I}}_{t}=\mathrm{Pois}\left({\hat{R}}_{t}\sum_{\tau =0}^{t}I_{t-\tau }\omega_{\tau }\right)$$
Here, ω denotes the serial-interval distribution, representing the time between successive cases in a transmission chain (i.e., the interval between infection and subsequent transmission) (this distribution follows a gamma function: $\omega \left(t\right)=\beta^{\alpha }t^{\alpha -1}\exp\left(-\beta t\right)$ where the parameters for COVID-19 have been estimated in [73] as α=1.87 and β=0.28).

Regarding the fatality rate, which serves as a measure of social and health damage, we refer to the study of the corrected case-fatality rate (λCFR) [71]. This metric adjusts for undiagnosed cases to provide a fatality estimation closer to the real infection-fatality rate. Using the collisional approach, the expected number of deaths can be expressed as follows:(8)$$E\left[D_{t}\right]=\frac{1}{\lambda }{\hat{I}}_{t}$$
Here, λ is influenced by various factors, including virus mutations, precautionary protocols, seasonal and environmental elements, pharmaceutical interventions, and the age distribution of individuals affected by the disease.

Using a methodology similar to the epidemiological-mobility approach, as in Proposition 3, we can imagine creating a quantitative relationship between macroeconomic costs and an entropy-based measure of collective human mobility. This involves assessing the implications of changes in human mobility patterns through a hypothetical macro-economic indicator C=f(ω,L), where L encompasses various mobility variables like movement trends, population density, and societal interaction patterns. Conversely, the variable ω represents a purely economic factor, capturing the specificity (elasticity) of the economic indicator and its sensitivity to changes in mobility and interactions within region-specific demographic, social, and economic contexts. However, due to the complexity of the subject, our empirical exercise represents only a qualitative and preliminary investigation for such a mathematical foundation.

In conclusion, we focused our research study on establishing a robust and quantitative description of population mobility, laying the groundwork for further (and more rigorous) studies that can connect socioeconomic indicators to mobility patterns. An exhaustive explanation of the empirical results goes beyond the purpose of the work and could lead to an interesting discussion on behavioral and transportation economics to better understand how spatial cognition shapes mobility patterns [74,75,76].

## 4. Final Remarks

Regarding patterns of mobility, entropy serves as a valuable metric for quantifying the level of disorder or randomness within human movement behaviors. As individuals traverse various locations and engage in different activities, the entropy of their mobility patterns provides insight into the diversity and unpredictability of their movements. Higher entropy values suggest greater variability and less predictability in mobility patterns, whereas lower entropy values indicate more structured and predictable behaviors. Moreover, entropy-based metrics are proposed as potential candidates for macroscopic connections between human mobility patterns and economic activity. Consequently, by analyzing entropy in mobility patterns, researchers can discern the underlying trends and dynamics, facilitating a deeper understanding of human behavior and informing decision-making processes in various domains such as urban planning, transportation, and public health.

This work shows that entropy, as a measure of the diversity of mobility, exhibits a stronger relationship with certain economic indicators, as also seen in [23]. The economic and social shock induced by the COVID-19 pandemic has significantly altered both individual and organizational perspectives on work and occupations, leading to shifts in occupational supply and demand dynamics, as well as changes in attitudes toward remote work. Consequently, our analysis of mobility regularities offers practical insights into how the COVID-19 pandemic has impacted human mobility and economic variables. For instance, lockdown measures have instigated sequential economic adjustments in response to shocks to productive capacity (supply shocks) and/or final demand (demand shocks) [24,77,78,79]. As a result, the epidemic has concurrently affected both the supply and demand sides of affected economies, primarily through reductions in mobility, increases in physical distancing, and implementation of stay-at-home confinement policies. These policy interventions have repercussions such as altering labor availability across sectors, constraining output allocation to the final demand and other sectors, influencing demand for non-labor inputs, and potentially creating bottlenecks in downstream sector production. Therefore, emphasizing entropy as a measure of mobility regularity, rather than solely focusing on mobility itself, can offer a more accurate predictor of the direct and indirect effects of an epidemic phenomenon on a region’s economy.

In addition, our theoretical framework can provide a more reliable estimate of perceived population density through a weight factor, which takes into account both regularities and the intensity of movements. We explored the relationship between mobility, entropy, and various economic variables. Understanding this correlation could be valuable for developing region-specific mitigation policies that effectively balance epidemic control and economic stability.

While we do not claim to offer an exhaustive model for mobility patterns and population behaviors, we view our work as a starting point for quantitatively representing the connection between human movement trends and regularities with demographic and economic indicators. The significance of our contribution lies in the fact that entropy encapsulates various aspects of social movement trends and has the potential to serve as a straightforward mobility indicator for policymakers. In future work, we will aim to expand the model to incorporate more detailed individual behavior information, such as travel trajectories and time spent in specific locations. Achieving this will require overcoming our model’s assumption that individuals’ movements are uncorrelated, thereby enhancing the reliability of describing dynamic socioeconomic phenomena driven by mobility trends.

## Figures and Tables

**Figure 1 entropy-26-00398-f001:**
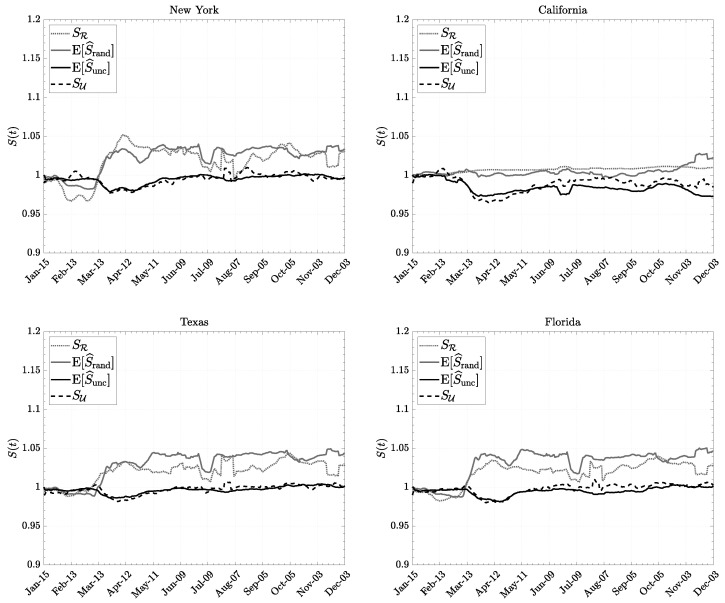
Entropies of individuals’ movements in 2020 for New York, California, Texas, and Florida. The time series have been rescaled by their values at the beginning of the observation period. Data courtesy of Camber Systems [46] and U.S. Census Bureau [49].

**Figure 2 entropy-26-00398-f002:**
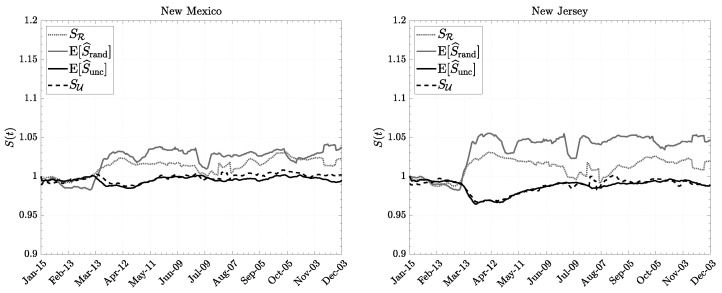
Comparison between the various implementations of the entropy of individuals’ movements. Data courtesy of Camber Systems [46] and U.S. Census Bureau [49] for New Mexico (a state with an evenly distributed population) and New Jersey (a state with an unevenly distributed population).

**Figure 3 entropy-26-00398-f003:**
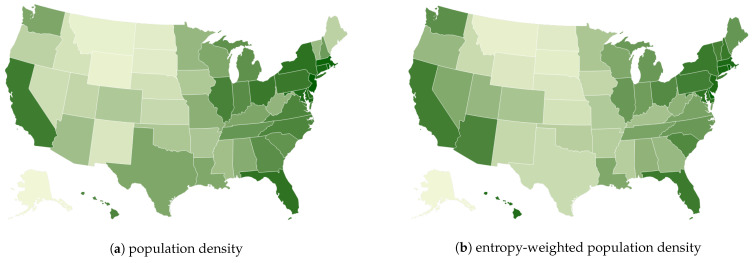
Comparison between the standard population density: people per unit area (**a**) and the population density weighed by the Shannon entropy correction factor (**b**). The intensity of green represents the value of the density, darker color means higher density.

**Figure 4 entropy-26-00398-f004:**
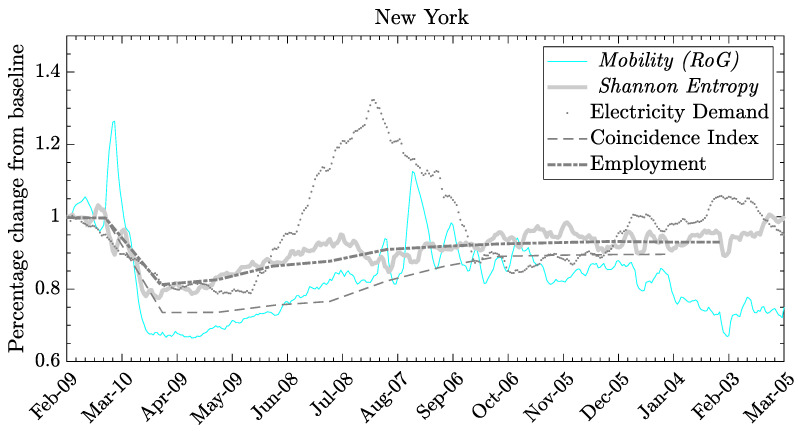
The economic indicators are reported monthly. Data courtesy of Camber Systems [46], U.S. Bureau of Labor Statistics of Labor Statistics [70], Federal Reserve Bank of Philadelphia [55], and Electricity demand EIA [54], for mobility, labor, economic, and energy data, respectively.

**Figure 5 entropy-26-00398-f005:**
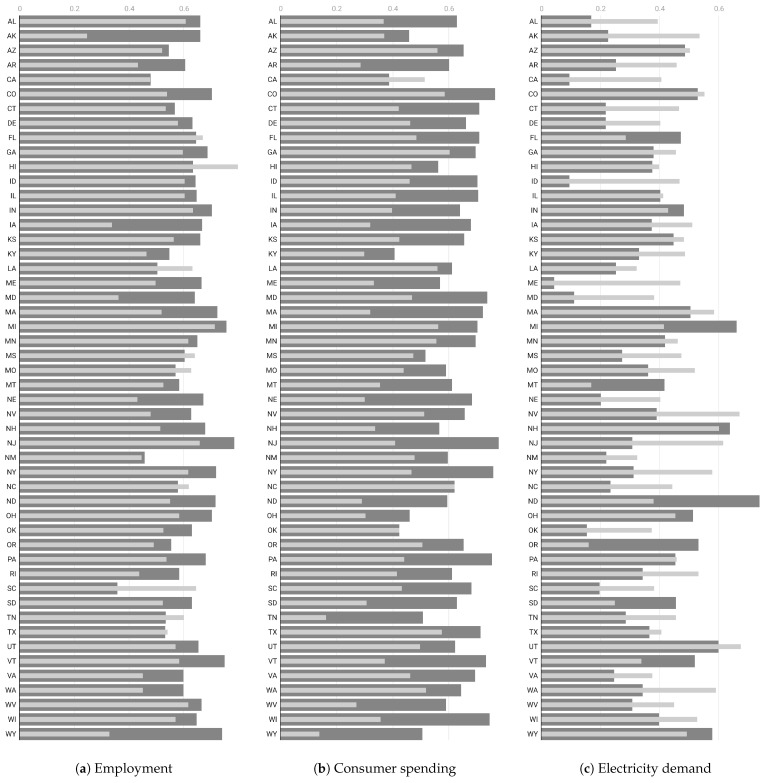
Correlation coefficient for each state between each economic indicator and the RoG mobility (light thin bars) and uncorrelated entropy (dark thick bars) for three different economic indicators: (**a**) employment rate, (**b**) consumer spending, and (**c**) energy consumption. Longer bars indicate a stronger correlation between the time series.

**Figure 6 entropy-26-00398-f006:**
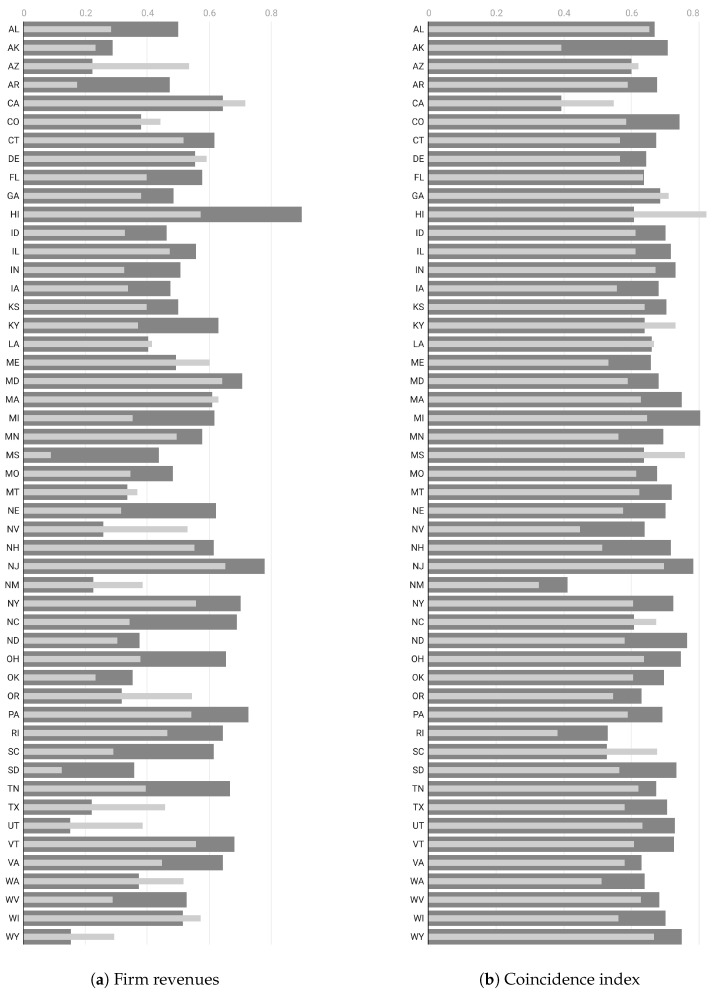
Correlation coefficient for each state between each economic indicator and the RoG mobility (light thin bars) and uncorrelated entropy (dark thick bars) for two different economic indicators: (**a**) firms’ revenues and (**b**) coincidence index. Longer bars indicate a stronger correlation between the time series.

**Table 1 entropy-26-00398-t001:** Mobility data structure.

State FIPS	Day	Mean Entropy	Std Entropy	Mean *ROG* (m)	Std *ROG* (m)	Device Count
48	15 January 2020	29.225	11.540	43,961.618	99,679.671	916,762
48	16 January 2020	28.479	11.400	44,261.272	103,866.417	957,096
⋮	⋮	⋮	⋮	⋮	⋮	⋮
48	13 March 2021	28.057	11.216	41,653.951	100,544.112	920,325

**Table 3 entropy-26-00398-t003:** State population density estimates for the year 2020.

State	Population Density (*d*)			
	Census [49]	Entropy-Weighted	Adjusted [49]	PWD-M [50]
New Jersey	466.3	474.0	785.4	1713.7
Rhode Island	395.6	1403.7	455.3	1403.3
Massachusetts	341.2	524.9	552.4	1275.1
Connecticut	284.3	688.3	371.4	916.8
Maryland	240.5	228.1	453.4	1580.6
Delaware	193.0	1061.5	259.8	977.0
New York	159.4	67.5	1323.0	3132.5
Florida	154.6	57.0	286.4	1345.3
Pennsylvania	110.5	43.1	276.0	980.6
Ohio	110.5	31.7	228.2	935.5
California	97.9	43.9	329.1	2873.6
Illinois	88.1	22.0	410.6	1676.6
Hawaii	85.1	350.7	250.6	2342.3
Virginia	83.5	17.4	286.8	1152.1
North Carolina	83.3	21.7	150.6	516.8
Indiana	72.6	19.8	138.1	824.6
Georgia	71.3	11.7	204.4	710.8
Michigan	68.2	21.6	201.3	857.4
South Carolina	66.1	33.0	97.5	483.7
Tennessee	63.9	16.8	123.7	513.1
New Hampshire	58.6	117.4	95.4	307.8
Washington	44.2	28.3	129.5	1489.4
Kentucky	43.7	9.5	91.3	474.9
Texas	42.9	4.8	245.4	1626.6
Louisiana	41.5	15.1	97.2	646.2
Wisconsin	41.5	15.0	112.5	849.3
Alabama	37.4	12.8	64.2	346.1
Missouri	34.5	7.7	113.9	853.2
West Virginia	28.8	13.1	44.8	220.4
Minnesota	27.3	8.5	128.9	981.6
Vermont	26.1	41.4	34.1	124.5
Arizona	24.7	37.6	76.0	1887.7
Mississippi	24.5	7.1	37.3	243.3
Arkansas	22.4	7.1	40.8	372.5
Oklahoma	22.3	6.9	73.2	736.2
Iowa	21.8	5.6	43.7	816.1
Colorado	21.5	9.0	121.2	1783.1
Oregon	17.0	11.9	76.9	1639.1
Maine	16.8	23.4	37.4	134.8
Utah	15.1	12.0	135.2	1907.0
Kansas	13.8	3.5	64.9	1032.3
Nevada	10.8	14.7	64.4	3016.4
Nebraska	9.7	2.7	71.0	1246.8
Idaho	8.3	4.7	35.1	1137.8
New Mexico	6.7	5.4	24.9	1058.6
South Dakota	4.5	1.8	12.5	677.2
North Dakota	4.3	2.1	9.6	878.1
Montana	2.8	1.4	6.9	515.4
Wyoming	2.3	2.4	3.3	694.3
Alaska	0.5	0.5	5.6	687.6

**Table 4 entropy-26-00398-t004:** Correlation coefficient among all 50 U.S. states. Median values of the correlation coefficient and their 10th and 90th percentiles.

*Indicator*	(RoG) Mobility	*C.I.*	(Shannon) Entropy	*C.I.*
Employment	*0.55*	[0.43, 0.64]	** *0.64* **	[0.53, 0.72]
Consumer Spending	*0.43*	[0.29, 0.56]	** *0.65* **	[0.48, 0.74]
Electricity Demand	** *0.46* **	[0.32, 0.59]	*0.36*	[0.16, 0.56]
Firms Revenue	* **0.51** *	[0.24, 0.69]	*0.34*	[0.26, 0.60]
Coincident index	*0.61*	[0.51, 0.69]	* **0.69** *	[0.61, 0.75]

## Data Availability

The mobility datasets presented in this article are not readily available because they were accessible only for a limited time during the COVID-19 pandemic. Requests to access the datasets should be directed to Camber System [46]. The demographic data presented in the study are openly available via the Census Bureau and World Pop Hub at [50,52]. Finally, economic data presented in this study are openly available in the Economic Tracker at [53], the U.S. Energy Information Administration [54], and the Federal Reserve Bank [55].

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
