# Peer review of "On an Aggregated Estimate for Human Mobility Regularities through Movement Trends and Population Density"

_entropy, 2024, doi:10.3390/e26050398_

Round 1

Reviewer 1 Report

Comments and Suggestions for Authors

This article introduces an analytical framework that interprets individual measures of entropy-based mobility derived from mobile phone data. The authors explore and analyze two widely recognized entropy metrics: random entropy and uncorrelated Shannon entropy. These metrics are estimated through collective variables of human mobility, including movement trends and population density. By employing a collisional model, the authors establish statistical relationships between entropy measures and mobility variables. Furthermore, the research addresses three primary objectives: firstly, validating the model; secondly, exploring correlations between aggregated mobility and entropy measures in comparison to five economic indicators; and finally, demonstrating the utility of entropy measures. Specifically, the authors provide an effective population density estimate that offers a more realistic understanding of social interactions. This estimation takes into account both movement regularities and intensity, utilizing real-time data analysis conducted during the peak period of the COVID-19 pandemic.

This work is interesting, but there are several flaws in the methodology.

Major revision

Entropy has been used to model daily cases of COVID-19, basic reproduction number, etc. These examples can be found in literature. The authors should have included this background of how entropy is useful in other domains.

I was expecting the authors to give a background about different methodology that can be used to simulate mobility, such as gravity model, radiation models, etc.

What can the authors say about mobility trend during the weekend as they consider time series data. Is there any difference between the weekday and weekend mobility trends?

It would have been nice to see the correlation between the method presented by authors on mobility and the cases or deaths due to COVID-19 pandemic. The authors only shows a nexus between mobility and socio-economic factors.

Minor revision

Covid-19 should be COVID-19 throughout the manuscript 

There are several grammatical errors. Such can be found in lines 193 and many others.

Comments on the Quality of English Language

Minor editing of English language required.

Author Response

We would like to thank the Reviewer for their insightful comments that allowed us to really improve the original version of the paper. Changes in the revised version are highlighted in red. Additionally, we provide below an answer to each comment raised.

This article introduces an analytical framework that interprets individual measures of entropy-based mobility derived from mobile phone data. The authors explore and analyze two widely recognized entropy metrics: random entropy and uncorrelated Shannon entropy. These metrics are estimated through collective variables of human mobility, including movement trends and population density. By employing a collisional model, the authors establish statistical relationships between entropy measures and mobility variables. Furthermore, the research addresses three primary objectives: firstly, validating the model; secondly, exploring correlations between aggregated mobility and entropy measures in comparison to five economic indicators; and finally, demonstrating the utility of entropy measures. Specifically, the authors provide an effective population density estimate that offers a more realistic understanding of social interactions. This estimation takes into account both movement regularities and intensity, utilizing real-time data analysis conducted during the peak period of the COVID-19 pandemic. 

This work is interesting, but there are several flaws in the methodology. 

Major revision 

  1. Entropy has been used to model daily cases of COVID-19, basic reproduction number, etc. These examples can be found in literature. The authors should have included this background of how entropy is useful in other domains.
    We appreciate the suggestion to expand the literature review to provide a more comprehensive framework for the topic addressed in our article in lines 22-28.
  2. I was expecting the authors to give a background about different methodology that can be used to simulate mobility, such as gravity model, radiation models, etc. 
    We agree it would be appropriate to include some literature about human mobility models beyond the one introduced in our article. For that reason, we added the lines 34-38. 
  3. What can the authors say about mobility trend during the weekend as they consider time series data. Is there any difference between the weekday and weekend mobility trends?
    In lines 215-230, we have included a discussion addressing the reviewer's point regarding weekly patterns. We have provided a brief discussion out of some evidence from our data, comparing the periods before and after COVID-19 restrictions. We believe this is an intriguing question that requires a more comprehensive statistical analysis to fully understand these phenomena. We intend to explore this further in future research endeavors, as outlined in the discussion section of the article. 
  4. It would have been nice to see the correlation between the method presented by authors on mobility and the cases or deaths due to COVID-19 pandemic. The authors only shows a nexus between mobility and socio-economic factors. 
    Following the reviewer's comment, we have delved deeper into the implications of Proposition 3 to establish an analytical framework based on our model for estimating daily cases and deaths resulting from the COVID-19 pandemic. To this end, we have included an additional paragraph in the paper, located at lines 370-384. 

    Minor revision 

    Covid-19 should be COVID-19 throughout the manuscript  

    There are several grammatical errors. Such can be found in lines 193 and many others. 

    We have conducted a thorough grammatical analysis of the text and addressed any errors we identified. 

Reviewer 2 Report

Comments and Suggestions for Authors

1. It would be nice to have a table of the summary descriptive statistics of the data used in the study.

2. The length of the period of the data used should be explained.

3. The use of US data should be justified. It should add some explanation for picking the US as a case study.

4. Some literature on studies identifying how the COVID-19 pandemic affected human mobility and how changes in human mobility affected the economic variables used in this study should be referred to and discussed.

5. I understand the paper's focus is more on methodological development but it would be nice to have more discussions on how the study result might contribute to understanding the practical implications of how the COVID-19 pandemic affected human mobility and economic variables investigated in this study.

Comments on the Quality of English Language

Overall English is fine but the use of the word COVID-19 should be consistent throughout the paper. I recommend using all capitalized letters for COVID.

Author Response

We would like to thank the Reviewer for their insightful comments that allowed us to really improve the original version of the paper. Changes in the revised version are highlighted in red. Additionally, we provide below an answer to each comment raised. 

  1. It would be nice to have a table of the summary descriptive statistics of the data used in the study.  

We agree that for the sake of completeness it is better to describe the data in more detail. To this scope, we have added a paragraph and a table where we explain the meaning of the Mobility data we have used. As regards the economic data, we explained that in section 3.3. 

2. The length of the period of the data used should be explained. 

We have added this information to the text as in point 1. 

3. The use of US data should be justified. It should add some explanation for picking the US as a case study. 

Essentially, the reason is that these databases were freely accessible and directly assessed the entropy metrics in which we were interested.  Another reason is that we have already analyzed that country in our previous published works.  It would be interesting to apply our analysis to other countries.  We have added motivations in lines 41-43 and lines 154-155. 

4. Some literature on studies identifying how the COVID-19 pandemic affected human mobility and how changes in human mobility affected the economic variables used in this study should be referred to and discussed.  

Thank you for the suggestion, we have added references and discussion about what pointed by the reviewer in lines 65-75 in the introduction. 

5. I understand the paper's focus is more on methodological development, but it would be nice to have more discussions on how the study result might contribute to understanding the practical implications of how the COVID-19 pandemic affected human mobility and economic variables investigated in this study.  

In the final remarks, we have incorporated a discussion on how studies on human mobility patterns, such as ours, can aid in comprehending the impact of Covid-19 on economic variables. We have inserted this discussion in the final remarks section, specifically at lines 416-432. 

Comments on the Quality of English Language  Overall English is fine but the use of the word COVID-19 should be consistent throughout the paper. I recommend using all capitalized letters for COVID. 

We did that, thank you

Round 2

Reviewer 1 Report

Comments and Suggestions for Authors

The authors have addressed my concerns.